# Effects of Baru Almond Oil (*Dipteryx alata Vog.*) Treatment on Thrombotic Processes, Platelet Aggregation, and Vascular Function in Aorta Arteries

**DOI:** 10.3390/nu14102098

**Published:** 2022-05-18

**Authors:** Cristiane Cosmo Silva-Luis, José Luiz de Brito Alves, Júlio César Pinheiro Lúcio de Oliveira, José Alixandre de Sousa Luis, Islania Giselia Albuquerque Araújo, Josean Fechine Tavares, Yuri Mangueira do Nascimento, Lorena Soares Bezerra, Fátima de Lourdes Assunção Araújo de Azevedo, Marianna Vieira Sobral, Vivianne Mendes Mangueira, Isac Almeida de Medeiros, Robson Cavalcante Veras

**Affiliations:** 1Health Sciences Center, Department of Nutrition, Campus I, Federal University of Paraíba, João Pessoa 58059-900, PB, Brazil; criscosmosilva@hotmail.com (C.C.S.-L.); lorena.sbezerra@gmail.com (L.S.B.); 2Health Sciences Center, Department of Pharmaceutical Sciences, Campus I, Federal University of Paraíba, João Pessoa 58059-900, PB, Brazil; juliopinheiro23@hotmail.com (J.C.P.L.d.O.); jose.alixandre@professor.ufcg.edu.br (J.A.d.S.L.); islania.giselia@academico.ufpb.br (I.G.A.A.); josean@ltf.ufpb.br (J.F.T.); yurimangueira@ltf.ufpb.br (Y.M.d.N.); fatimazevedo@gmail.com (F.d.L.A.A.d.A.); mariannasobral@gmail.com (M.V.S.); viviannemangueira@gmail.com (V.M.M.); isacmedeiros@uol.com.br (I.A.d.M.); drrobveras@gmail.com (R.C.V.)

**Keywords:** *Dipteryx alata Vog.*, *Fabaceae*, tocopherols, thrombosis, platelet aggregation, vascular response

## Abstract

Background: This study assessed the effects of Baru (*Dipteryx alata Vog.*) almond oil supplementation on vascular function, platelet aggregation, and thrombus formation in aorta arteries of Wistar rats. Methods: Male Wistar rats were allocated into three groups. The control group (*n* = 6), a Baru group receiving Baru almond oil at 7.2 mL/kg/day (BG 7.2 mL/kg, *n* = 6), and (iii) a Baru group receiving Baru almond oil at 14.4 mL/kg/day (BG 14.4 mL/kg, *n* = 6). Baru oil was administered for ten days. Platelet aggregation, thrombus formation, vascular function, and reactive oxygen species production were evaluated at the end of treatment. Results: Baru oil supplementation reduced platelet aggregation (*p* < 0.05) and the production of the superoxide anion radical in platelets (*p* < 0.05). Additionally, Baru oil supplementation exerted an antithrombotic effect (*p* < 0.05) and improved the vascular function of aorta arteries (*p* < 0.05). Conclusion: The findings showed that Baru oil reduced platelet aggregation, reactive oxygen species production, and improved vascular function, suggesting it to be a functional oil with great potential to act as a novel product for preventing and treating cardiovascular disease.

## 1. Introduction

Baru almond (*Dipteryx alata Vog.*, *Fabaceae* family) is an emerging nut from the central-western area of the Brazilian Savanna [1]. Baru nut displays a high nutritional value, notably rich in unsaturated fatty acid (such as oleic, linolenic gadoleic, and erucic), fiber, protein, alpha-tocopherol, and several polyphenols’ compounds [1,2,3]. Such nutritional characteristics have conferred on the Baru almond an essential source of natural antioxidants with health-promoting effects and promising beneficial effects on chronic diseases [4].

An early study demonstrated that daily treatment with Baru oil at a dose of 1 g/kg/day for 15 weeks attenuated lipid peroxidation and drastically reduced liver damage in dyslipidemic rats [5]. Additionally, clinical findings have found beneficial effects of Baru almonds or Baru oil supplementation on adiposity, lipid profile, inflammation, and antioxidant properties in subjects with cardiometabolic disorders [4,6,7,8]. Despite promising pre- and clinical findings of Baru oil supplementation against cardiometabolic diseases, the effects of Baru almond oil on vascular function, platelet aggregation, and thrombus formation remain to be elucidated.

Thrombosis is characterized by forming an aggregation of platelets, fibrin, and red blood cells in the vascular system. Arterial thrombosis is part of the spectrum of cardiovascular diseases (CVDs) with high severity [9], and this constitutes one of the leading causes of death associated with the occurrence of ischemic stroke and acute ischemic myocardial infarction [10]. In the physiopathology of thrombosis, reactive oxygen species (ROS) production within the activated platelets has received particular attention. An imbalance in the antioxidant system leads to platelet hyperaggregability, increasing the risk of thrombus formation [11]. In this way, antioxidant compounds should be an essential strategy to combat oxidative stress and exert a platelet antiaggregant and antithrombotic effect. Early findings have reported an antithrombotic effect and decreased ADP-induced platelet aggregation after oral administration of tocopherols [12,13,14].

Considering the beneficial cardiometabolic effects and tocopherol concentration in Baru almond oil, we have evaluated the effects of Baru almond oil (*Dipteryx alata Vog.*) on vascular function, platelet aggregation, superoxide anion production, and thrombus formation in aorta arteries of Wistar rats.

## 2. Materials and Methods

### 2.1. Experimental Design

Male Wistar rats (*Rattus norvergicus*, 100 days of age) were used in this study. The rats were maintained in the Animal Production Unit of the Institute for the Research on Drugs and Medications (IPeFarM) of the Federal University of Paraiba (UFPB) in collective polypropylene cages (3 animals/cage) with controlled temperature (22 ± 1 °C), humidity (50–55%) and light-dark cycle (12 h), receiving water and diet ad libitum. The procedures followed the National Council for Control of Animal Experimentation (CONCEA) and International Principles for Biomedical Research. The experimental protocols were approved by an Institutional Animal Care Committee (CEUA-UFPB protocol number # 128/2016).

Rats were grouped into: (i) a control group (*n* = 6), receiving phosphate-buffered saline (PBS) as placebo; (ii) a Baru group receiving Baru almond oil at 7.2 mL/kg/day (BG 7.2 mL/kg, *n* = 6); and (iii) a Baru group receiving Baru almond oil at 14.4 mL/kg/day (BG 14.4 mL/kg, *n* = 6). The placebo or Baru almond oil was administered daily with oral gavage for ten days. Baru oil (Lot B43) was purchased from Ybá Óleos Puros Company (Alto Paraíso de Goiás, Goiania, Brazil).

### 2.2. Determination of the Serum Tocopherol Concentration

Serum tocopherol was determined by a high-performance liquid chromatography system equipped with an LC-20AT quaternary solvent pumping module, SIL-20A autoinjector, DGU-20A degassing system, RF-20A detector, and CBM-20A controller. A CLC-ODS (M) Shim-pack column and an aG-ODS-4 Shim-pack pre-column (Shimadzu, Japan) for tocopherol determination were used [3]. The identification and quantification of α-, (β + γ)-and δ-tocopherol in serum was performed using standard curves prepared with external standards corresponding to α-tocopherol, y = 310,236x − 1 × 10^−6^, r^2^: 0.9951; (β + γ)-tocopherol, y = 584,174x − 2 × 10^−6^, r^2^: 0.9993; and δ-tocopherol, y = 652,752x − 2 × 10^−6^, r^2^: 0.9975 [15].

### 2.3. Induction of Thrombosis and Monitoring of Blood Flow

Anesthetized rats were coupled with an X1 ultrasound probe and a data acquisition system (T106 Transonic Systems Inc., New York, NY, USA). The carotid artery flow was recorded using the LabChart software version 8.0 (AD Instruments, Bella Vista, NSW, Australia). After 10 min of baseline recording, a thrombus was induced by inserting a piece of Whatman No. 1 paper embedded with 5 μL of FeCl_3_ solution [16] into the ventral surface of the carotid artery [17,18,19]. The occlusion time is when the flow reaches the value of zero and remains for 30 s [20]. Subsequently, a segment of the carotid artery was dissected, and the thrombus was opened and removed for the immediate measurement of the weight of the thrombus (wet weight) and after drying for 24 h at 37 °C (dry weight) [20].

### 2.4. Obtaining Whole Blood and the Preparation of Platelets

Blood was collected from the inferior vena cava using sterile heparinized syringes. To determine platelet aggregation by light transmission and evaluate the production of O_2_^•−^ in platelet-rich plasma (PRP), the blood was deposited in siliconized plastic tubes and centrifuged at 120 g for 10 min to obtain the PRP. Platelet-poor plasma (PPP) was obtained after second centrifugation at 2300× *g* for 15 min [21,22].

### 2.5. Measurement of Platelet Aggregation and Superoxide Anions (O2^•−^) Production

PRP samples (adjusted to 2 × 10^7^ platelets/mL) were stimulated with the adenosine 5′-diphosphate (ADP, 16.7 µM, Sigma Aldrich A2754, Burlington, MA, USA) or phorbol 12-myristate 13-acetate (PMA, 100 µM, Sigma Aldrich 79346, Burlington, MA, USA). For ADP stimulation, the recording time was 10 min, and for PMA, the recording time was 15 min (AgreGO, Sao Paulo, Brazil). The degree of aggregation was expressed as a percentage of the maximum light transmission obtained with PRP [23].

DHE probe (500 µM, Sigma Aldrich D7008, Burlington, MA, USA) was added to 200 µL of the platelet suspension, protected from light, and placed at 37 °C for 30 min. After washout, the fluorescence was evaluated with a flow cytometer FACS CANTO II equipped with a 15-mW argon laser, λ = 488 nm (BD, Santa Monica, CA, USA). A total of 10,000 events were acquired in the FITC channel (564–606 nm) [21,22], followed by analysis using the DIVA software 6.0 (BD, Santa Monica, CA, USA).

### 2.6. Vascular Reactivity

The thoracic aorta artery was isolated and rings (1–2 mm) mounted in an organ bath (10 mL), suspended vertically, and attached to a force transducer (MLT020, AD Instruments, Bella Vista, NSW, Australia). The tissues were maintained in Krebs solution at 37 °C and aerated with 95% O_2_ and 5% CO_2_ (Carbogen, White Martins, Brazil). All the rings were subjected to a basal tension of approximately 1.0 g during a stabilization period of 60 min, exchanging the nutrient solution every 15 min [24]. After stabilization and viability verification, vascular endothelium was considered intact when the aortic rings showed ACh-induced relaxation greater than 70% after Phe (1 µM, Sigma Aldrich PHR1017, Burlington, MA, USA) contraction. Endothelium removal was confirmed by a relaxation of less than 10% [24]. Next, isolated concentration–response curves for Phe (10^−9^ − 10^−5^ M), ACh (10^−9^ − 10^−5^ M, TCI America A008425G, Portland, OR, USA) and SNP (10^−12^ − 10^−5^ M, Sigma Aldrich 71778, Burlington, MA, USA) were obtained. The last two were constructed shortly after contraction induced by Phe (1 μM) reached the tonic phase.

### 2.7. Statistical Analysis

The results were described as mean ± standard deviation for parametric data or median (maximum-minimum) for non-parametric data. A Kolmogorov–Smirnov test was used to assess data normality. Parametric variables were analyzed with one-way ANOVA and a Tukey post hoc test. Non-parametric variables were compared with a Kruskal–Wallis test with a Dunn’s post hoc test. The E_max_ and pD2 values for the concentration–response curves were obtained through nonlinear regressions. Statistical analysis was carried out with Prism 6 software (GraphPad Software 6, San Diego, CA, USA). A *p*-value of < 0.05 was considered significant.

## 3. Results

### 3.1. Tocopherol Measurements in Baru Oil

A typical chromatogram of the separation of tocopherols in Baru oil showed three well-defined peaks identified by external standards as δ-, (β + γ)- and α-tocopherol (Appendix A). The results revealed the presence of 3.3 μg/mL of δ-Tocopherol, 18.3 μg/mL of (β + γ)-tocopherol, and 7.3 μg/mL of α-tocopherol, totaling 28.9 μg/mL.

### 3.2. Effects of Baru Oil Treatment Plasma Body Weight and α-Tocopherol Concentration

Body weight and plasmatic α-tocopherol concentration (F = 1.81, *p* = 0.19) were similar among control and rats receiving Baru oil (Figure 1A,B, respectively).

### 3.3. Effects of Baru Oil Treatment on Thrombus Formation, Platelets Aggregation, and Superoxide Anion Scavenging

Baru oil treatment at a dose of 14.4 mL/kg reduced thrombus establishment induced by FeCl3. Rats receiving Baru oil treatment at 14.4 mL/kg exhibited increased occlusion time when compared to the control and Baru group receiving 7.2 mL/kg (F = 14.9, *p* = 0.0003, Figure 2A). In addition, wet thrombus weight (F = 9.33, *p* = 0.002) and dry thrombus weight (F = 7.24, *p* = 0.006) were significantly reduced in rats receiving Baru oil at 14.4 mL/kg when compared to the control group (Figure 2B,C).

Regarding ADP-induced platelet aggregation, oil Baru treatment at 14.4 mL/kg decreased platelet aggregation compared to the control group (F = 4.97, *p* = 0.02, Figure 2D). However, Baru oil treatment did not alter platelet aggregation in PMA stimulation (F = 0.10, *p* = 0.90, Figure 2E).

Lastly, Baru oil treatment at 7.2 mL/kg and 14.4 mL/kg reduced superoxide anion in aggregated platelets compared to the control group (F = 16.4, *p* = 0.0002, Figure 2F).

### 3.4. Effects of Baru Oil Treatment on Vascular Reactivity

The vascular responses assessed in aortic artery rings with functional endothelium are illustrated in Figure 3A, while vascular responses assessed in aortic artery rings without functional endothelium are shown in Figure 3B. Baru oil treatment with 7.2 mL/kg and 14.4 mL/kg reduced significantly (*p* < 0.05) maximum effect (Emax) of phenylephrine in aorta rings with and without endothelium (Figure 3A,B).

The vascular relaxation assessed with functional endothelium using acetylcholine is demonstrated in Figure 4A. Baru oil treatments did not alter the Emax response and potency indicated by pD2 (Figure 4A). The vascular relaxations assessed in aorta rings without functional endothelium using sodium nitroprusside are demonstrated in Figure 4B. Baru oil treatments did not modify the Emax response, but the potency indicated by pD2 was increased in rats receiving Baru oil treatment at 14.4 mL/kg (Figure 4B).

## 4. Discussion

The present study demonstrated that the administration of Baru almond oil (*Dipteryx alata Vog.*) decreased thrombus establishment, platelet aggregation, and superoxide anion production in rats. Additionally, Baru almond oil treatment for ten days effectively improved vascular function in aorta arteries, suggesting a great potential to be helpful in the treatment and prevention of cardiovascular disorders.

Regarding tocopherol contents, we have found 28.9 mg/kg of tocopherols in Baru oil, In fact, the tocopherol content is low when compared to other almond oils, such as Fournat, Ferraduel, and Ferragnes [25]. Although it is not possible to state the reason for this difference, a recent study demonstrated that the conditions of Baru oil extraction could interfere with tocopherol contents [26]. Despite this, it is reasonable to highlight other nutritional qualities of the Baru oil, notably rich in oleic, linoleic, and linolenic fatty acids and poor in saturated fatty acid palmitate [1,2,27].

Oxidative stress is conceptualized as “an imbalance between oxidants and antioxidants in favor of the oxidants, leading to a redox signaling and molecular damage [28]. Early evidence has reported that a disturbance in the production and removal of the reactive species has been associated with platelet activation and thrombotic processes [29,30]. Platelet NADPH oxidase seems to be the primary source of platelet reactive oxidant species and platelet hyperreactivity [29,30]. In this way, therapeutic strategies that protect against oxidative stress and reduce platelet reactive oxygen species should represent a window of opportunity to prevent platelet activation and thrombotic processes. Baru almond oil reduced superoxide anion in aggregated platelets. Considering that NOX2 is an isoform of NADPH oxidase expressed by the leading producer of ROS in platelets, further studies will be needed to elucidate molecular mechanisms by which Baru oil reduces superoxide anion in aggregated platelets.

ADP can interact with P2Y1 and P2Y12 receptors on platelets and induce platelet aggregation and thrombus formation [31]. Early findings reported that natural products, such as olive oil, omega-3 polyunsaturated fatty acids, cocoa, and phenolic compounds, could partially inhibit ADP-induced platelet aggregation [32,33]. Here, for the first time, we have demonstrated that Baru almond oil treatment for ten days reduced ROS production, decreased approximately 31% of ADP-induced platelet aggregation, and decreased thrombotic processes in rats, suggesting it to be effective in reducing platelet activation and exerting benefits in the thrombosis processes.

Increased oxidative stress has been reported in the development of endothelial dysfunction. In addition, platelet hyperactivity associated with endothelial dysfunction has been described as having a critical role in the pathogenesis of atherosclerotic vascular complications [34]. In our study, Baru almond oil administration improved vascular function by reducing vasoconstrictor properties in aorta rings. Although the underlying mechanisms by which Baru almond oil administration improves vascular function have not been explored in the present study, our results can help to include Baru oil consumption in the cardiovascular approach.

Randomized, placebo-controlled trials have demonstrated that Baru almond oil supplementation improved serum lipid parameters in mildly hypercholesterolemic subjects [8], increased glutathione peroxidase antioxidant enzyme activity in overweight and obese women [7], and decreased ultra-sensitive C-reactive protein concentration in patients with chronic kidney disease under hemodialysis treatment [4]. However, further studies will be needed to assess whether Baru almond oil supplementation effectively improves endothelial function in subjects with cardiovascular diseases.

The lack of hematological, biochemical, and histopathological analyses of blood and organ samples could be described as a limitation of this study.

In summary, our findings revealed that Baru almond oil (*Dipteryx alata Vog.*) might be a safe and promising strategy to reduce platelet aggregation, thrombotic process, and improve endothelial function (Figure 5).

## Figures and Tables

**Figure 1 nutrients-14-02098-f001:**
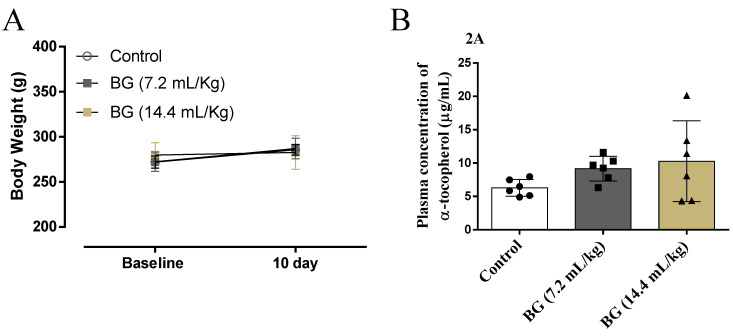
Assessment of body weight (**A**) and plasma concentration of α-tocopherol (**B**) in rats. Data are presented as mean ± standard deviation and analyzed by ANOVA one-way test with Tukey as a post-hoc test.

**Figure 2 nutrients-14-02098-f002:**
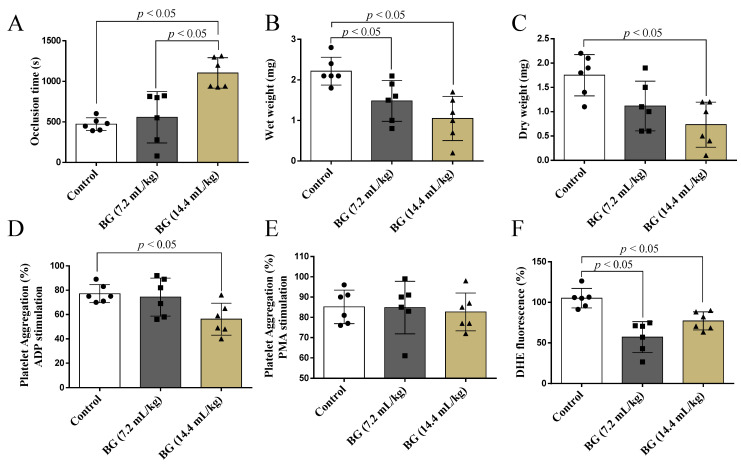
Effects of Baru almond oil (*Dipteryx alata Vog.*) on platelet aggregation, thrombus formation, and reactive oxygen species in male rats. Assessment of occlusion time (**A**), wet weight of thrombus (**B**), dry weight of thrombus (**C**), ADP-induced platelet aggregation (**D**), PMA-induced platelet aggregation (**E**), and DHE fluoresce in platelet (**F**) in male rats. Groups: control group (control), Baru group receiving 7.2 mL/kg of Baru almond oil (BG, 7.2 mL/kg), and Baru group receiving 14.4 mL/kg of Baru almond oil (BG, 14.4 mL/kg). Data are presented as mean ± standard deviation and analyzed by ANOVA one-way test with Tukey as a post-hoc test.

**Figure 3 nutrients-14-02098-f003:**
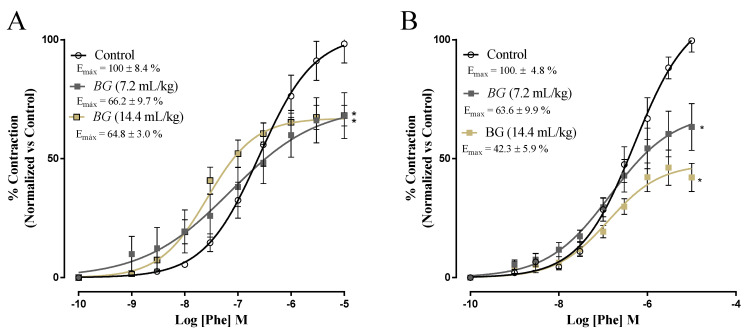
Effects of Baru almond oil (*Dipteryx alata Vog.*) on endothelial function in male rats. Evaluation of vascular reactivity in aorta artery rings. Concentration–response curve to phenylephrine with (**A**) and without functional endothelium (**B**). * *p*-value < 0.05 vs. control group.

**Figure 4 nutrients-14-02098-f004:**
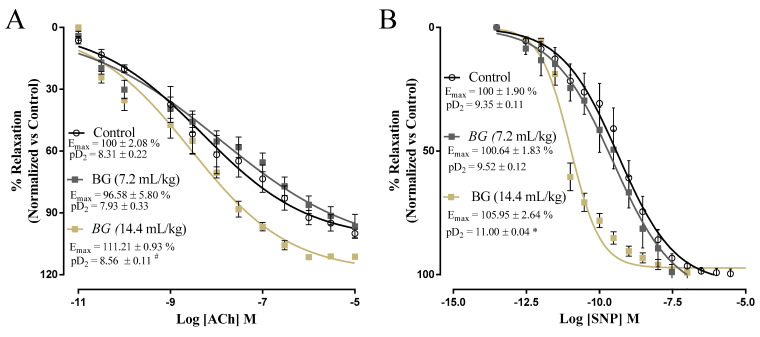
Effects of Baru almond oil (*Dipteryx alata Vog.*) on endothelial function in male rats. Evaluation of vascular reactivity in aorta artery rings. Concentration–response curve for acetylcholine (Ach, **A**) in rings with functional endothelium and dose–response curve for sodium nitroprusside (SNP, **B**) in rings without functional endothelium. * *p*-value < 0.05 vs. control group. # *p*-value < 0.05 vs. BG 7.2 mL/kg group.

**Figure 5 nutrients-14-02098-f005:**
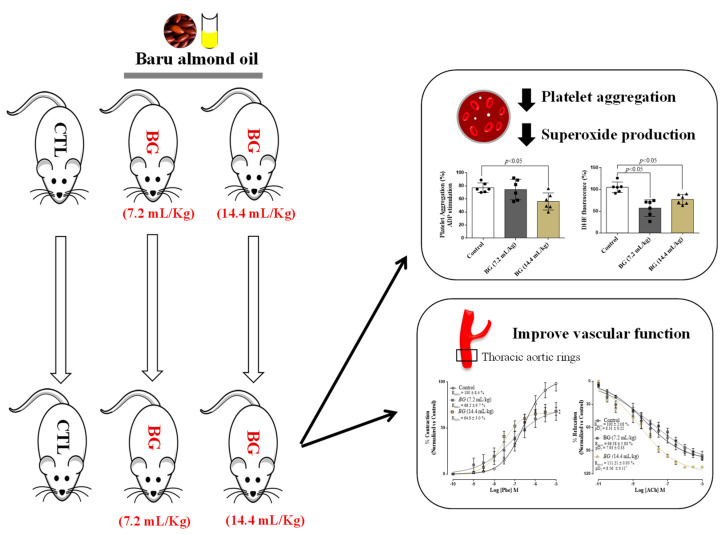
Schematic drawing showing the effect of the Baru almond oil administration on platelet aggregation, superoxide anion generation, and vascular function in rats. CTL (control), BG (Baru group).

## Data Availability

The data that support the findings of this study are available on request from the corresponding author. The data are not publicly available due to privacy or ethical restrictions.

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
