# Peer review of "Effects of Baru Almond Oil (Dipteryx alata Vog.) Treatment on Thrombotic Processes, Platelet Aggregation, and Vascular Function in Aorta Arteries"

_nutrients, 2022, doi:10.3390/nu14102098_

Round 1
Reviewer 1 Report
Summary:
The authors of the manuscript “Effects of Baru almond oil (Dipteryx alata Vog.) treatment on
thrombotic processes, platelet aggregation, and vascular function in aorta arteries” have reported the promising effects of Baru oil in terms of vascular function, oxidative stress, and thrombus formation. there are a few significant concerns in the current version of the manuscript
- The introduction is not self-explanatory and needs to be improved.
- On page 1, line 20, the authors have mentioned that ten rats were used; however, I see that only n=6-8 were reported in the methodology.
- The authors need to share the exact number of rats used in the study per group, but not the range of (6-8)
- What was the source of Baru almond oil? Did you purchase it from a commercial vendor? If so, please share the catalog no. for the same.
- Mention the units of liquids in milliliters (ml) but not in milligrams
- In the figure legends, mention the no. of samples used for the analysis and p-value as well
- In the methodology section, include catalog no. of all the reagents used study
- When you are sure that there is no toxicity below 5000 mg/kg, what was the rationale for doing this acute toxicity study? Did you also check for the biomarkers of vital organs in the serum and histopathology of the heart, kidney, and liver?
- On page 2, line 78, the authors mentioned, “Daily placebo or Baru almond oil was administered with oral gavage for ten days” what a placebo is in this study?
Author Response
Reviewer #1
The authors of the manuscript “Effects of Baru almond oil (Dipteryx alata Vog.) treatment on thrombotic processes, platelet aggregation, and vascular function in aorta arteries” have reported the promising effects of Baru oil in terms of vascular function, oxidative stress, and thrombus formation. there are a few significant concerns in the current version of the manuscript
Author response: We would like to thank reviewer 1 for recognizing the relevance of our work and the opportunity to improve our manuscript. We have made the adjustments proposed by the reviewer, which contributed substantially to the improvement of the manuscript.
- The introduction is not self-explanatory and needs to be improved.
Author response: We agree with reviewer 1. In the revised version, we’ve improved the introduction.
- On page 1, line 20, the authors have mentioned that ten rats were used; however, I see that only n=6-8 were reported in the methodology.
Author response: Thank you reviewer for this observation. Sorry for this mistake. We’ve corrected this information in the revised version.
- The authors need to share the exact number of rats used in the study per group, but not the range of (6-8)
Author response: Thank you reviewer for this observation. We’ve added the exact number of rats in the revised version.
- What was the source of Baru almond oil? Did you purchase it from a commercial vendor? If so, please share the catalog no. for the same.
Author response: Thank you reviewer 1 for this observation. Baru oil was purchased from a commercial vendor. In the revised version, we’ve added in materials and methods: “Baru oil (Lot B43) was purchased from Ybá Óleos Puros Company (Alto Paraíso de Goiás, Goiania, Brazil)”.
- Mention the units of liquids in milliliters (ml) but not in milligrams
Author response: We’ve corrected the units in the revised version.
- In the figure legends, mention the no. of samples used for the analysis and p-value as well
Author response: We’ve changed the figures and added p-value<0.05 for statical differences.
- In the methodology section, include catalog no. of all the reagents used study
Author response: We’ve added the catalog of reagents in the revised version.
- When you are sure that there is no toxicity below 5000 mg/kg, what was the rationale for doing this acute toxicity study? Did you also check for the biomarkers of vital organs in the serum and histopathology of the heart, kidney, and liver?
Author response: Considering that acute toxicity was determined only in terms of their general behavior, adverse clinical signs, and mortality for the first hour, after four hours, and subsequently for 14 d at regular intervals for 12 h and no biomarkers of vital organs in the serum and histopathology of the heart, kidney, and live, we’ve decided to remove this result of the manuscript. In addition, we’ve added in the discussion section a potential limitation of the study.
- On page 2, line 78, the authors mentioned, “Daily placebo or Baru almond oil was administered with oral gavage for ten days” what a placebo is in this study?
Author response: We’ve added this information in the revised version. PBS was used as placebo.
Reviewer 2 Report
In the present work, the authors use a rat model to study the possible effects of the integration of Baru almond oil (Dipteryx alata Vog) on thrombotic processes and vascular function. They would demonstrate that Baru almond oil treatment can improve vascular function and prevent arterial thrombosis inducing a significant decrease in thrombus formation, platelet aggregation, and ROS production. The study and its conclusions are interesting; however, I have some questions and suggestion.
- In figure 2 Panel D-F the plots indicate that a different number of animals were used and compared. What is the reason?
- Figure 2, panel D: Usually, high platelet aggregation correlates with increase of ROS production, but the authors showed contradictory results. For example, In Figure 2D the authors observed a reduction in platelet aggregation after supplementation of Baru (14.4 mg/kg). Instead, in Figure 2F there is a significant reduction in superoxide anion production in the group treated with Baru 7.2. mg / kg but not in the 14.4 mg / kg group. How can the authors explain this result? It would seem that 7.2. mg /kg is the best functional concentration.
- Many authors performed experiments for long time (4 weeks) and using higher doses of Baru oil compared with this. How did the authors choose time and doses of oil administration?
- How did you choose the ADP concentration? What happens if the platelets are stimulated with collagen and or thrombin?
- In addition to the DHE fluorescence analysis, considering that Nox2 is an isoform of NADPH oxidase expressed by the main producer of ROS in platelets, did the authors evaluate the activity of NOX2 and production of H2O2? It could be interesting to evaluate the molecular mechanisms such as evaluation of p47 and/or PLA2 phosphorylation, upstream and downstream pathways of NOX2, respectively.
- The representation of the statistical results on the graphs is confusing. Please, specify the meaning of * or #, thus of the p-value in the figure legend.
- Is the complete composition of the oil known? Since the tocopherol contents are low compared to other almond oils (example: Fournat, Ferraduel, Ferragnes), do you think that there is another component with protective function? What could it be?
- A modified endothelial function could be supported by the evaluation of different biomarkers, such as NO (a vasodilator molecule) production.
- The animal used in this study are healthy and without pathological condition. In fact, in the methods section any pathological condition was described. Nevertheless, the results showed that the almond oil acts also in physiological condition. Why? What’s the possible clinical impacts and meaning?
- Why did control rats show high levels of oxidative stress and platelet aggregation? Explain it.
- Moreover, I think, it would be better to compare rats fed with a high-fat diet versus normal diet to support the authors' thesis.
- Finally, I suggest adding a schematic figure to better describe and complete the proposed work.
Author Response
Reviewer #2
In the present work, the authors use a rat model to study the possible effects of the integration of Baru almond oil (Dipteryx alata Vog) on thrombotic processes and vascular function. They would demonstrate that Baru almond oil treatment can improve vascular function and prevent arterial thrombosis inducing a significant decrease in thrombus formation, platelet aggregation, and ROS production. The study and its conclusions are interesting; however, I have some questions and suggestion.
Author response: We would like to thank reviewer 2 for recognizing the relevance of our work and the opportunity to improve our manuscript. We have made the adjustments proposed by the reviewer, which contributed substantially to the improvement of the manuscript.
- In figure 2 Panel D-F the plots indicate that a different number of animals were used and compared. What is the reason?
Author response: Thank you reviewer 2 for this observation. In fact, we started the experiment with n=8 per group. However, we had structural problems (lack of energy, water, and other technical problems) during the execution of the vascular reactivity protocols. Considering this, we decided to use only rats that passed through all protocols. We’ve corrected the figure in the revised version.
- Figure 2, panel D: Usually, high platelet aggregation correlates with increase of ROS production, but the authors showed contradictory results. For example, In Figure 2D the authors observed a reduction in platelet aggregation after supplementation of Baru (14.4 mg/kg). Instead, in Figure 2F there is a significant reduction in superoxide anion production in the group treated with Baru 7.2. mg / kg but not in the 14.4 mg / kg group. How can the authors explain this result? It would seem that 7.2. mg /kg is the best functional concentration.
Author response: Sorry reviewer 2. We’ve found a reduced platelet aggregation and superoxide anion production using 14.4 ml/kg of Baru oil. In fact, we don't see it as a contradictory results.
- Many authors performed experiments for long time (4 weeks) and using higher doses of Baru oil compared with this. How did the authors choose time and doses of oil administration?
Author response: To determine the oil doses (volume of oil to be administered to rats) we standardized it by its tocopherol (Vitamin E) content. So, the first step was the quantification of tocopherols by HPLC. Subsequently, we verified the Dietary Reference Intakes (DRI) of the IOM39 of Vitamin E for a 70 kg adult, the Dietary Recommendation is 15 mg/day for humans of both sexes. From this, we calculate the volume of oil that corresponds to this Vitamin E content.
The tocopherol contents found in 1 mL of D. alata Vog. were: 3.3 μg of δ-tocopherol, 18.3 μg of (β+γ)-tocopherol and 7.3 μg of α-tocopherol, totalizing 28.9 μg/ml. Dividing 15 per 70, we got a dose of 0.21 mg/kg of tocopherol. Considering that each ml of Baru almond oil contains 0.0289 mg of tocopherol, we proposed the doses of 7.2 ml/kg and 14.4 ml/kg of Baru almond oil.
- How did you choose the ADP concentration? What happens if the platelets are stimulated with collagen and or thrombin?
Author response: The choice of ADP concentration used in this study was based on standard protocols performed in our laboratory. At 16.7uM, greater stability of platelet aggregation was observed. Collagen and thrombin are also used in other protocols of platelet aggregation.
- In addition to the DHE fluorescence analysis, considering that Nox2 is an isoform of NADPH oxidase expressed by the main producer of ROS in platelets, did the authors evaluate the activity of NOX2 and production of H2O2? It could be interesting to evaluate the molecular mechanisms such as evaluation of p47 and/or PLA2 phosphorylation, upstream and downstream pathways of NOX2, respectively.
Author response: We agree with reviewer 2. However, NOX2 and H2O2 production was not evaluated in the present study. In lines 233-236, we’ve added: “Baru almond oil reduced superoxide anion in aggregated platelets. Considering that NOX2 is an isoform of NADPH oxidase expressed by the main producer of ROS in plate-lets, further studies will be needed to elucidate molecular mechanisms by which Baru oil reduces superoxide anion in aggregated platelets”.
- The representation of the statistical results on the graphs is confusing. Please, specify the meaning of * or #, thus of the p-value in the figure legend.
Author response: Thank you reviewer for your observation. In revised version, we’ve removed “*” and “#” and added p-value<0.05 for statistic difference.
- Is the complete composition of the oil known? Since the tocopherol contents are low compared to other almond oils (example: Fournat, Ferraduel, Ferragnes), do you think that there is another component with protective function? What could it be?
Author response: Thank you reviewer for his/her question. We agree with reviewer 2 that tocopherol contents are low compared to other almond oils. In the revised version, we have added the following sentence:
“Regarding tocopherol contents, we have found 28,9 mg/kg of tocopherols in Baru oil, In fact, the tocopherol content is low when compared to other almond oils, such as Fournat, Ferraduel, Ferragnes (26). Although it is not possible to state the reason for this difference, a recent study demonstrated that the conditions of Baru oil extraction could interfere with tocopherol contents (27). Despite this, it is reasonable to highlight other nutritional qualities of the Baru oil, notably rich in oleic, linoleic, and linolenic fatty acids and poor in saturated fatty acid palmitate (1,2,28).
- A modified endothelial function could be supported by the evaluation of different biomarkers, such as NO (a vasodilator molecule) production.
Author response: Thanks so much reviewer 2 for his/her suggestion. Future studies will be carried out to investigate NO pathways.
- The animal used in this study are healthy and without pathological condition. In fact, in the methods section any pathological condition was described. Nevertheless, the results showed that the almond oil acts also in physiological condition. Why? What’s the possible clinical impacts and meaning?
Author response: Our aim was to assess the effect of Nutrient Baru almond oil. with our result, it is not possible to affirm any possible clinical impact. However, our results bring new scientific information demonstrating that Baru almond oil display antiplatelet properties and improves vascular function, something also reported with olive oil. If such effects occur in pathological conditions remain to be elucidated.
- Why did control rats show high levels of oxidative stress and platelet aggregation? Explain it.
Author response: Sorry reviewer 2, but the control rats did not show high lives of oxidative stress and platelet aggregation. Our results showed that Baru oil reduced platelet aggregation and superoxide anion. We investigated specifically the effect of nutrient Baru oil on superoxide anion and platelet aggregation. If Baru oil has an effect on oxidative stress conditions remains to be elucidated.
- Moreover, I think, it would be better to compare rats fed with a high-fat diet versus normal diet to support the authors' thesis.
Author response: Thank you reviewer for his/her suggestion. In fact, our aim was to test the function of the nutrient Baru almond oil. Certainly, future studies will be carried out to evaluate this proposal.
- Finally, I suggest adding a schematic figure to better describe and complete the proposed work.
Author response: Thank you. A schematic figure was added in revised version. Please, see figure 5.
Round 2
Reviewer 1 Report
The investigative group of the manuscript “Effects of Baru almond oil (Dipteryx alata Vog.) treatment on thrombotic processes, platelet aggregation, and vascular function in aorta arteries” justified all my concerns legitimately in the revised version. I hope this manuscript will draw the attention of both basic and clinical researchers.